# Analytical Solution of Thermo–Mechanical Properties of Functionally Graded Materials by Asymptotic Homogenization Method

**DOI:** 10.3390/ma15093073

**Published:** 2022-04-23

**Authors:** Dan Chen, Lisheng Liu, Liangliang Chu, Qiwen Liu

**Affiliations:** 1Department of Mechanics and Engineering Structure, Wuhan University of Technology, Wuhan 430070, China; 292562@whut.edu.cn; 2Hubei Key Laboratory of Theory and Application of Advanced Materials Mechanics, Wuhan University of Technology, Wuhan 430070, China; qiwen_liu@whut.edu.cn

**Keywords:** functionally graded materials, asymptotic homogenization method, effective properties, thermo–mechanical coupling

## Abstract

In this work, a general mathematical model for functionally graded heterogeneous equilibrium boundary value problems is considered. A methodology to find the local problems and the effective properties of functionally graded materials (FGMs) with generalized periodicity is presented, using the asymptotic homogenization method (AHM). The present models consist of the matrix metal Mo and the reinforced phase ceramic ZrC, the constituent ratios and the property gradation profiles of which can be described by the designed volume fraction. Firstly, a new threshold segmentation method is proposed to construct the gradient structure of the FGMs, which lays the groundwork for the subsequent research on the properties of materials. Further, a study of FGMs varied along a certain direction and the influence of the varied constituents and graded structures in the behavior of heterogeneous structures are investigated by the AHM. Consequently, the closed–form formulas for the effective thermo–mechanical coupling tensors are obtained, based on the solutions of local problems of FGMs with the periodic boundary conditions. These formulas provide information for the understanding of the traditional homogenized structure, and the results also be verified the correctness by the Mori–Tanaka method and AHM numerical solution. The results show that the designed structure profiles have great influence on the effective properties of the present inhomogeneous heterogeneous models. This research will be of great reference significance for the future material optimization design.

## 1. Introduction

The growing demand and optimization of industrial applications reveal the limitations of traditional materials. The development of functionally graded materials (FGMs) solves these limitations to a large extent by integrating mutually exclusive properties [1,2,3]. FGMs are a kind of advanced composite materials and show a local characteristic dependence on the spatial distribution of their constituent phases [4,5], which were first designed for the Japanese space shuttle project in 1983 to reduce the thermal stress caused by the high temperature of the metal and ceramic interface. This advanced heterogeneous composite improves the thermal shock resistance of the material along the gradient direction [6,7]. The superior material properties of FGMs had been widely applied in different areas, such as in electronic packaging, automobile and aerospace industries [8]. Many studies had been done on the fabrication, application, and mechanical properties [9,10,11,12] of FGMs. How to predict and determine accurately the properties of such inhomogeneous FGMs is vital research problem for obtaining the thermo–mechanical ability under the conditions of some extreme environment [13,14]. However, the properties of FGMs depend not only on the constituent materials, but also on their microstructure. The complexity of the microstructure of FGMs, i.e., the content of components, distribution of inclusions and so on, makes the prediction of their equivalent thermo–mechanical properties very untoward [15].

In recent years, a great deal of research has focused its attention on the prediction of the effective properties of composite materials. Theoretical boundary methods include Hashin–Shtrikman bounds [16,17,18,19] and Budiansky bounds [20], but these methods do not consider the complexity of microstructure. For composites, it is very vital to understand their accurate properties after the microstructure is determined. Based on Eshelby’s equivalent inclusion theory [21], combined with the average stress, self–consistent method [22], generalized self–consistent method [23] and Mori–Tanaka method [24,25,26] are proposed to predict the effective properties of composites. For example, Tran et al. [25] derived the solution of Eshelby’s spherical heterogeneous problem by using the Mori–Tanaka method to predict the effective properties of gradient composites with spherical inclusions. However, because Eshelby’s equivalent inclusion theory is based on ellipsoid, these methods are initially only applicable to composites with ellipsoidal inclusions. For composites with complex shapes, the finite element analysis method (FEM) with periodic boundary conditions can be used [27,28]. Kundalwal et al. [27] studied the effective properties of fiber–reinforced composites by using the FEM, and calculated the effective elastic constants of the materials. Zhang et al. [28] studied the multi field properties of electromagnetic thermoelastic composites by using the FEM based on micromechanics, and obtained the effective thermal expansion coefficient and other parameters of intelligent composites under periodic boundary conditions. However, these methods lack strict mathematical framework to clarify the relationship between the effective properties and the material layout. Recently, based on strict mathematical derivation, a novel homogenization method is proposed, asymptotic homogenization method (AHM), and shows good accuracy compared with experiments [29,30,31]. The method of AHM can be incorporated into the coupling between the micro and macro behaviors of heterogeneous materials [32,33,34,35,36].

It can be seen from the existing literature that AHM has been widely used in the prediction and analysis of the effective properties of composites, with good calculation results. In terms of prediction of the effective mechanical properties [37,38,39,40], for example, Santana et al. [38] predicted the effective constitutive coefficient tensor of the composite by using two–scale AHM considering the effect of debonding between fiber and matrix. The comparison of homogenization and non–homogenization numerical models shows that almost all effective coefficients have good convergence. Nasirov et al. [39] further used the three–scale formulas of AHM to predict Young’s modulus of fiber from micro scale to mesoscale and from mesoscale to macro scale, which is in good agreement with the tensile test. The study of multi physical field coupling has further developed the application of AHM, especially the prediction of effective thermo–mechanical properties [41,42,43,44,45]. Muhammad et al. [42] established AHM for three–scale composite analysis considering the thermomechanical effect, and compared the Young’s modulus and Poisson’s ratio obtained from the analysis with the experimental results. The results show that the two methods have good consistency in Young’s modulus. Zhao et al. [43] studied the thermo–mechanical properties of particle–filled polymers using AHM. The results show that the numerical results of progressive homogenization method are in good agreement with the experimental results. For periodic composites, the progressive homogenization method is more reasonable and accurate than the representative volume element method. Dirichlet formulation overestimates the effective tensor and Neumann formulation underestimates the effective tensor. AHM can also be used to study multi physical field coupling such as mechano–chemical coupling [46], thermo–electric coupling [47], and magneto–electric coupling [48]. However, the research on the effective properties of FGMs by AHM has not been reported.

Because AHM is based on strict mathematical framework, it finally leads to a complex implementation, which is mainly related to the complex microstructure of heterogeneous materials [49]. In general, the relationship between macro and micro structure is realized through the concept of the representative volume unit (RVE) [50,51,52]. RVE needs to characterize accurately the microstructure of composites. Tian et al. [53,54] proposed the RVE generation method of spatially randomly distributed fibers in combination with the random sequential absorption (RSA) technique, and then Tian et al. [55] used RSA technique to generate the RVE of composites containing cylindrical inclusions. At the same time, Dutra et al. [15] believe that the steps of AHM using numerical simulation are not simple, such as applying the first boundary condition and periodic boundary condition. When Ramos et al. [56] studied two–phase parallel fiber reinforced periodic viscoelastic composites, a simple closed–form formula for the effective properties of square and hexagonal element composites was obtained by using AHM. Based on the elastic viscoelastic correspondence principle and assuming that there are perfect contact conditions at each split interface, the local problems and global viscoelastic properties are obtained in explicit form. Compared with the traditional Maxwell and Kelvin models, its effectiveness is verified. It can be seen that the closed–form formula of effective performance not only has good accuracy, but also does not need to build a complex microstructure model for numerical solution, which can greatly reduce the demand for computers and can be better used in practice. Therefore, when studying the effective thermo–mechanical properties of FGMs, it is very important to deduce the closed–form formula of effective thermo–mechanical properties by AHM on the premise of ensuring the accuracy of calculation results.

In this study, a methodology to solve the local problems and the effective properties of FGMs with generalized periodicity is presented base on the method of AHM. For FGMs with special distribution, i.e., power–law distribution with two parameters, the regulating effects of different gradient models on FGMs thermo–mechanical performance are analyzed. Gaussian random field method and our new threshold segmentation algorithm are used to establish a two–phase random gradient structure that obeys the power–law distribution of two parameters, so as to show the distribution characteristics of different gradient model components. The theoretical closed–form formulas for calculating the performance of FGMs are derived from the gradual homogenization method and its generalized periodicity. For the thermo–mechanical coupling field, these properties include elastic tensor’s components, thermal expansion tensor’s components and thermal conduction tensor’s components. The Mori–Tanaka method and numerical solution of AHM in literature have been used to verify the correctness of the present method. The quantitative relationship between the gradient structure and thermo–mechanical properties of FGMs and the resulting analysis is an important part of our research.

## 2. The Establishment of the Gradient Microstructure

In this section, the Gaussian random field method combined with a new threshold segmentation algorithm is mainly considered to construct the FGM model. For various random media, various construction methods based on finite morphological and statistical information extracted from 2D/3D data are proposed, including random–set method [57], Gaussian random field method [58,59] etc. The random–set method can be used for composites with spherical or polygonal inclusions, but for inclusions with arbitrary shape, the calculation is complex and difficult to realize. In the Gaussian random field method, the field–field correlation function is used to construct the Gaussian field, and the horizontal cutting is used to obtain the two–phase micro model. According to [59], the 2D original gray random distribution microstructure model of heterogeneous materials can be obtained. Generally, in order to obtain the two–phase microstructure model, it is necessary to use an image segmentation algorithm to separate the target phase and background matrix. The determination of the cut–off value in the segmentation algorithm depends on whether the volume fraction of the target phase of the obtained two–phase model is consistent with the target model.

When a cut–off value is selected in the whole gray model, a uniform two–phase model with a certain volume fraction will be obtained. Obviously, this cannot establish the FGM model with gradient distribution of the volume fraction of the component phase. In order to construct the gradient model, we propose a new threshold segmentation algorithm, that is, the whole model is divided into several regions, the number of regions depends on the number of layers of the gradient model itself, and different cut–off values are selected in each block according to the required volume fraction.

It is assumed that the thickness of FGM models studied is *h*, which is composed of metal phase and ceramic phase. The properties of the studied model are assumed to be isotropic and homogeneous in the cross section while following a power law distribution with two parameters in the thickness direction i.e., *V_p_*(*z*) is the volume fraction of the component at the distance *z* from the bottom according to Ref. [60], which can be expressed as,
(1)$$V_{p}(z)=\left\{ \begin{array}{l} \frac{c_{0}}{2}{\left(1+\frac{2z}{h}\right)}^{m},-\frac{h}{2}\leq z\leq 0\\ c_{0}\left[1-\frac{1}{2}{\left(1-\frac{2z}{h}\right)}^{m}\right],0\leq z\leq \frac{h}{2} \end{array}\right.$$
and the volume fraction of the component of the top surface satisfies,
(2)$$V_{b}(z)=1-V_{p}(z)$$
where *c*_0_ denotes the component ratio coefficient to adjust the constituent ratio of FGMs, 0 ≤ *c*_0_ ≤ 1.0; *m* is the gradient index of material property to describe the property gradation profiles of FGMs, i.e., *m* = 0 corresponds to the homogeneous traditional composition. The volume fraction *V_p_*(z) varies with the structure of FGMs (*c*_0_, *m*), as shown Figure 1. Once the composition of the material composition ratio is determined, the current structure can only be adjusted by the structural design method through the gradient index (Figure 1b). Given the existence of two parameters, component ratio coefficient and gradient coefficient, this gradient structure is more scientific, which is beneficial to explore the influence of different volume component phase distribution on the thermo–mechanical properties of FGM structure.

According to the new threshold segmentation algorithm, the FGMs model with the volume fraction of component phase following Equations (1) and (2) in the thickness direction is constructed. The gradient model shows different gradient distributions according to the component ratio coefficient *c*_0_ and gradient index *m*, as shown in Figure 2 and Figure 3. It is observed that the dark phase and the light phase of the model show the opposite change trend in the longitudinal direction, i.e., they represent two different component phases, respectively. The volume proportion of the light phase from bottom to top is increasing, showing a gradient change.

In Figure 2, the component ratio coefficient *c*_0_ = 1.0 of the gradient models have different gradient indices *m*. For different *m*, the models have little difference near *z*/*h* = 0, but the farther away from this position, the greater the difference. With the increase of *m*, the closer the position where the light phase begins to appear to the top surface, the narrower the position area where the two–phase components form the interactive structure. When the gradient index *m* = 1.0, the gradient model with different component ratio coefficient *c*_0_ is constructed as shown in Figure 3. There is only the dark phase when *c*_0_ = 0. The volume fraction of the light phase increased gradually from bottom to top in the thickness direction, (*c*_0_ ≠ 0), but there was no drastic change trend. At the same time, the two–phase components almost form an interactive structure in the whole thickness direction. With the increase of *c*_0_, the proportion of the overall volume of the light phase increases continuously. When *c*_0_ = 1.0, only the light phase is on the top surface.

It should be noted that what are constructed here are not the actual microstructure, but the microstructure model similar to the actual FGMs. Strictly speaking, the gradient change of the gradient model is not continuous, but actually presents a stepped gradient change according to the region. In fact, the real FGMs is not a material with continuous gradient change in the ideal sense.

## 3. Asymptotic Homogenization Method

In this section, the local problems and effective properties of FGMs with generalized periodicity are derived and solved by AHM, and the theoretical closed calculation formulas for calculating the thermo–mechanical properties of FGMs are obtained, including effective elastic tensor’s components, effective thermal expansion tensor’s components and effective thermal conduction tensor’s components.

Let Ω ⊂ R^3^ be a three–dimensional open connected bounded domain with an infinitely smooth boundary ∂Ω. The equilibrium problem on a thermal mechanical coupling structure Ω is given by the kinetic equation and heat conduction equation,
(3)$$\frac{\partial \sigma_{ij}}{\partial x_{j}}=-f_{i}\;\mathrm{on}\;\Omega ,\;\mathrm{with}\;u=0\;\mathrm{on}\;\partial \Omega$$
(4)$$\frac{\partial q_{i}}{\partial x_{i}}=0\;\mathrm{on}\;\Omega ,\;\mathrm{with}\;u=0\;\mathrm{on}\;\partial \Omega$$
where, **σ** denotes the stress acting on the domain Ω; **f** is the body force; **x** is macroscopic (global) coordinate system; **q** is the thermal circulation vector.

Stress–strain relationship in linear thermomechanical problem can be expressed as follows,
(5)$$\delta_{ij}=C_{ijkl}(\varepsilon_{kl}-\varepsilon_{kl}^{t})$$
where, **C** is the four–order elasticity tensor of the material; **ε** is the total strain, and **ε^t^** denotes thermal strain.

The mechanical strain–displacement relationship and thermal strain be stated as, respectively,
(6)$$\varepsilon_{ij}=\frac{1}{2}\left(\frac{\partial u_{i}}{\partial x_{j}}+\frac{\partial u_{j}}{\partial x_{i}}\right)$$
(7)$$\varepsilon_{ij}^{t}=\alpha_{ij}\Delta T$$
where, **u** denotes the actual displacement; **α** is the second–order thermal expansion tensor; *T* is the temperature field, and ∆*T* = *T* − *T*_0_, *T*_0_ is the initial temperature. 

By substituting Equation (7) into Equation (5), and considering the symmetry of elastic tensor, the stress can be expressed as follows,
(8)$$\sigma_{ij}=C_{ijkl}\varepsilon_{kl}-\beta_{ij}\Delta T$$
where *β_ij_* = *C_ijkl_·α_kl_*. 

For the thermal circulation vector **q**,
(9)$$q_{i}=-\kappa_{ij}\frac{\partial T}{\partial x_{j}}$$
where **κ** is the second–order thermal conductivity tensor. 

These coefficients hold the following symmetry relationships, *C_ijkl_ = C_jikl_ = C_klij_*, *β_ij_ = β_ji_*. To complete our problem formulation, we compliment the boundary conditions with the following interphase conditions, ||*σ_ij_ n_j_*|| = 0, ||*q_i_n_i_*|| = 0, where ||A|| denotes a discontinuity in the value of A. 

### 3.1. Asymptotic Analysis and Model Development

It is apparent that the problem at hand is characterized by two scales: the macroscopic scale of order **x** which accounts for the variation of the dependent variables from one unit–cell to the next, and the microscopic scale of order **y** which accounts for periodicity. Appropriately, the first step in the asymptotic homogenization technique is the definition of a new microscopic variable *y_i_* as follows,
(10)$$y_{i}=\frac{x_{i}}{\varepsilon }$$
with a constant 0<ε<<1. Equation (10) represents the ratio of the length unit vector in the microscopic coordinates to the length unit vector in the macroscopic coordinates. 

And the partial derivative becomes,
(11)$$\nabla_{i}y=\frac{\partial }{\partial x_{i}}+\frac{1}{\varepsilon }\frac{\partial }{\partial y_{i}}$$

Then introduction of **y** necessitates the transformation, through the application of the chain rule, Equation (3) becomes the following Equation (12) and Equation (4) becomes the following Equation (13),
(12)$$\frac{\partial \sigma_{ij}(x,y)}{\partial x_{j}}+\frac{1}{\varepsilon }\frac{\partial \sigma_{ij}(x,y)}{\partial y_{j}}=-f_{i}\;\mathrm{on}\;\Omega ,\;\mathrm{with}\;u(x,y)=0\;\mathrm{on}\;\partial \Omega$$
(13)$$\frac{\partial q_{i}(x,y)}{\partial x_{i}}+\frac{1}{\varepsilon }\frac{\partial q_{i}(x,y)}{\partial y_{i}}=0\;\mathrm{on}\;\Omega ,\;\mathrm{with}\;u(x,y)=0\;\mathrm{on}\;\partial \Omega$$

And Equation (8) becomes the following Equation (14), Equation (9) becomes the following Equation (15),
(14)$$\sigma_{ij}(x,y)=\frac{1}{2}C_{ijkl}(y)\left\{\frac{\partial u_{k}(x,y)}{\partial x_{l}}+\frac{1}{\varepsilon }\frac{\partial u_{k}(x,y)}{\partial y_{l}}\right\}-\beta_{ij}(y)\Delta T(x,y)$$
(15)$$q_{i}(x,y)=-\kappa_{ij}(y)\frac{\partial T(x,y)}{\partial x_{j}}$$

The next step in the model development is to asymptotically expand the stress and displacement fields as well as the thermal circulation into infinite series in terms of integral powers of the small parameter *ε*. However, According to [61], the first and second order expansion terms of the AHM are necessary to obtain more micro information regarding composite structures for the thermo–mechanical problem. This means that the second order expansion term must be considered to ensure the accuracy of AHM for the thermo–mechanical problem. Moreover, according to [61], the first two order expansions are sufficiently accurate for most composite problems in general. Thus, those parameters are expanded depend on *ε*,

(1) Basic Expansions
(16)$$u_{i}(x,y)=u_{i}^{\left(0\right)}(x,y)+\varepsilon u_{i}^{\left(1\right)}(x,y)+\varepsilon^{2}u_{i}^{\left(2\right)}(x,y)+O(\varepsilon^{3})$$
(17)$$T(x,y)=T^{\left(0\right)}(x,y)+\varepsilon T^{\left(1\right)}(x,y)+\varepsilon^{2}T^{\left(2\right)}(x,y)+O(\varepsilon^{3})$$

(2) Derived Expansions
(18)$$\sigma_{ij}(x,y)=\sigma_{ij}^{\left(0\right)}(x,y)+\varepsilon \sigma_{ij}^{\left(1\right)}(x,y)+\varepsilon^{2}\sigma_{ij}^{\left(2\right)}(x,y)+O(\varepsilon^{3})$$
(19)$$q_{i}(x,y)=q_{i}^{\left(0\right)}(x,y)+\varepsilon q_{i}^{\left(1\right)}(x,y)+\varepsilon^{2}q_{i}^{\left(2\right)}(x,y)+O(\varepsilon^{3})$$

The first items $u_{i}^{\left(0\right)}(x,y)$, $T^{\left(0\right)}(x,y)$, $\sigma_{ij}^{\left(0\right)}(x,y)$, $q_{i}^{\left(0\right)}(x,y)$ in Equations (16)–(19) represent the physical quantities in macroscale; The second items $u_{i}^{\left(1\right)}(x,y)$, $T^{\left(1\right)}(x,y)$, $\sigma_{ij}^{\left(1\right)}(x,y)$, $q_{i}^{\left(1\right)}(x,y)$ are the microscale physical quantities; The items $u_{i}^{\left(n\right)}(x,y)$, $T^{\left(n\right)}(x,y)$, $\sigma_{ij}^{\left(n\right)}(x,y)$, $\sigma_{ij}^{\left(n\right)}(x,y)(n=2,3,\cdots )$ are the physical quantities in eventual smaller scales.

To obtain the asymptotic expansion for the mechanical stress and heat flux, substituting the asymptotic expansion Equations (16)–(19) into the constitutive equations Equations (14) and (15) and get,
(20)$$\sigma_{ij}^{\left(n\right)}=C_{ijkl}\left(\frac{\partial u_{k}^{\left(n\right)}}{\partial x_{l}}+\frac{\partial u_{k}^{\left(n+1\right)}}{\partial y_{l}}\right)-\beta_{ij}\Delta T^{\left(n\right)},(n=0,1,2\cdots )$$
(21)$$q_{i}^{\left(n\right)}=-\kappa_{ij}\left(\frac{\partial T^{\left(n\right)}}{\partial x_{j}}+\frac{\partial T^{\left(n+1\right)}}{\partial y_{j}}\right),(n=0,1,2\cdots )$$
where, $\Delta T^{\left(n\right)}(n=0,1,2\cdots )$ represents the change of temperature compared with the initial temperature at the different scales, similar to $T^{\left(n\right)}(n=0,1,2\cdots )$.

Taking the above relations Equations (18) and (19) into the governing Equations (12) and (13), one can get,
(22)$$\frac{\partial \sigma_{ij}^{\left(0\right)}}{\partial x_{j}}+\varepsilon \frac{\partial \sigma_{ij}^{\left(1\right)}}{\partial y_{j}}+\frac{1}{\varepsilon }\frac{\partial \sigma_{ij}^{\left(0\right)}}{\partial x_{j}}+\frac{\partial \sigma_{ij}^{\left(1\right)}}{\partial y_{j}}+\varepsilon^{2}\frac{\partial \sigma_{ij}^{\left(2\right)}}{\partial x_{j}}+\varepsilon \frac{\partial \sigma_{ij}^{\left(2\right)}}{\partial y_{j}}=-f_{i}$$
(23)$$\frac{\partial q_{i}^{\left(0\right)}}{\partial x_{j}}+\varepsilon \frac{\partial q_{i}^{\left(1\right)}}{\partial y_{j}}+\frac{1}{\varepsilon }\frac{\partial q_{i}^{\left(0\right)}}{\partial x_{j}}+\frac{\partial q_{i}^{\left(1\right)}}{\partial y_{j}}+\varepsilon^{2}\frac{\partial q_{i}^{\left(2\right)}}{\partial x_{j}}+\varepsilon \frac{\partial q_{i}^{\left(2\right)}}{\partial y_{j}}=0$$

Subsequently, equating like powers of *ε*, we obtain a series of differential equations.

For the terms of the stress field expansion,
(24)$$O(\varepsilon^{-1}):\frac{\partial \sigma_{ij}^{\left(0\right)}}{\partial y_{j}}=0$$
(25)$$O(\varepsilon^{0}):\frac{\partial \sigma_{ij}^{\left(0\right)}}{\partial x_{j}}+\frac{\partial \sigma_{ij}^{\left(1\right)}}{\partial y_{j}}=-f_{i}$$
(26)$$O(\varepsilon^{n}):\frac{\partial \sigma_{ij}^{\left(n\right)}}{\partial x_{j}}+\frac{\partial \sigma_{ij}^{\left(n+1\right)}}{\partial y_{j}}=0,(n=1,2,3\cdots )$$

For the terms of the thermal balance field expansion,
(27)$$O(\varepsilon^{-1}):\frac{\partial q_{k}^{\left(0\right)}}{\partial y_{k}}=0$$
(28)$$O(\varepsilon^{n}):\frac{\partial q_{k}^{\left(n\right)}}{\partial x_{k}}+\frac{\partial q_{k}^{\left(n+1\right)}}{\partial y_{k}}=0,(n=0,1,2\cdots )$$

### 3.2. Asymptotic Homogenization and Governing Equations

(1) For the displacement field problem

The determination of displacement field problem is Equations (24) and (25). We obtain $\sigma_{ij}^{\left(0\right)}$, $\sigma_{ij}^{\left(1\right)}$ from Equation (20),
(29)$$\sigma_{ij}^{\left(0\right)}=C_{ijkl}\left(\frac{\partial u_{k}^{\left(0\right)}}{\partial x_{l}}+\frac{\partial u_{k}^{\left(1\right)}}{\partial y_{l}}\right)-\beta_{ij}\Delta T^{\left(0\right)}$$
(30)$$\sigma_{ij}^{\left(1\right)}=C_{ijkl}\left(\frac{\partial u_{k}^{\left(1\right)}}{\partial x_{l}}+\frac{\partial u_{k}^{\left(2\right)}}{\partial y_{l}}\right)-\beta_{ij}\Delta T^{\left(1\right)}$$

Then substituting Equation (29) into Equation (24), we can get,
(31)$$\frac{\partial }{\partial y_{j}}\left\{C_{ijkl}\frac{\partial u_{k}^{\left(1\right)}}{\partial y_{l}}\right\}=\frac{\partial \beta_{ij}}{\partial y_{j}}\Delta T^{\left(0\right)}-\frac{\partial C_{ijkl}}{\partial y_{j}}\frac{\partial u_{k}^{\left(0\right)}}{\partial x_{l}}$$

The separation of variables on the right–hand side of the above equation prompts us to write down the solution of $u_{n}^{\left(1\right)}$ as,
(32)$$u_{n}^{\left(1\right)}(x,y)=V_{n}(x)+N_{n}^{kl}(y)\frac{\partial u_{k}^{\left(0\right)}(x)}{\partial x_{l}}+L_{n}(y)\Delta T^{\left(0\right)}(x)$$
where, *N^kl^*(**y**) and *L*(**y**) are the local functions of equilibrium problem to be solved, *V*(**x**) is the homogeneous part of the solution. 

Substituting Equation (32) into (31), the functions satisfy,
(33)$$\frac{\partial }{\partial y_{j}}\left\{C_{ijkl}\frac{\partial V_{n}(x)}{\partial y_{l}}\right\}=0$$
(34)$$\frac{\partial }{\partial y_{j}}\left\{C_{ijkl}(y)\frac{\partial N_{n}^{kl}(y)}{\partial y_{l}}\right\}=-\frac{\partial C_{ijkl}(y)}{\partial y_{j}}$$
(35)$$\frac{\partial }{\partial y_{j}}\left\{C_{ijkl}(y)\frac{\partial L_{n}(y)}{\partial y_{l}}\right\}=\frac{\partial \beta_{ij}(y)}{\partial y_{j}}$$

Substituting Equation (32) into (29), then,
(36)$$\sigma_{ij}^{\left(0\right)}=\left(C_{ijkl}(y)+C_{ijmn}(y)\frac{\partial N_{m}^{kl}(y)}{\partial y_{n}}\right)\frac{\partial u_{k}^{\left(0\right)}(x)}{\partial x_{l}}-\left(\beta_{ij}(y)-C_{ijmn}(y)\frac{\partial L_{m}(y)}{\partial y_{n}}\right)\Delta T^{\left(0\right)}(x)$$

Substituting Equation (36) into (25), and subsequently average over the volume of the unit–cell. Thus,
(37)$$\begin{array}{c} \frac{1}{\left|Y\right|}\int_{Y}\frac{\partial \sigma_{ij}^{\left(1\right)}(x,y)}{\partial y_{j}}dv+\frac{1}{\left|Y\right|}\int_{Y}\left(C_{ijkl}(y)+C_{ijmn}(y)\frac{\partial N_{m}^{kl}(y)}{\partial y_{n}}\right)dv\frac{\partial^{2}u_{k}^{\left(0\right)}(x)}{\partial x_{j}\partial x_{l}}\\ -\frac{1}{\left|Y\right|}\int_{Y}\left(\beta_{ij}(y)-C_{ijmn}(y)\frac{\partial L_{m}(y)}{\partial y_{n}}\right)dv\frac{\partial (\Delta T^{\left(0\right)}(x))}{\partial x_{j}}=-f_{i} \end{array}$$
where, *Y* is the dimension vector of the unit–cell, and |*Y*| is the volume of the unit–cell.

Keeping the periodicity of the associated functions in mind and remembering to treat *x_i_* as a parameter as far as integration with respect to **y** is concerned results in an expression of the form,
(38)$${\bar{C}}_{ijkl}\frac{\partial^{2}u_{k}^{\left(0\right)}(x)}{\partial x_{j}\partial x_{l}}-{\bar{\beta }}_{ij}\frac{\partial (\Delta T^{\left(0\right)}(x))}{\partial x_{j}}=-f_{i}$$With
(39)$${\bar{C}}_{ijkl}=\frac{1}{\left|Y\right|}\int_{Y}\left(C_{ijkl}(y)+C_{ijmn}(y)\frac{\partial N_{m}^{kl}(y)}{\partial y_{n}}\right)dv$$
(40)$${\bar{\beta }}_{ij}=\frac{1}{\left|Y\right|}\int_{Y}\left(\beta_{ij}(y)-C_{ijmn}(y)\frac{\partial L_{m}(y)}{\partial y_{n}}\right)dv$$

The first integral vanishes as a consequence of the periodicity of $\sigma_{ij}^{\left(1\right)}$ and following the application of the divergence theorem. 

(2) For the heat conduction problem

According to Equation (28), we can obtain,
(41)$$\frac{\partial q_{k}^{\left(0\right)}}{\partial x_{k}}+\frac{\partial q_{k}^{\left(1\right)}}{\partial y_{k}}=0$$

Thus, the heat conduction problem can be determined by Equations (27) and (41).

According to Equation (21), substituting and equating like powers of *ε*, we obtain, for $q_{k}^{\left(0\right)}$, $q_{k}^{\left(1\right)}$,
(42)$$q_{k}^{\left(0\right)}(x,y)=-\kappa_{ik}(y)\left(\frac{\partial T^{\left(0\right)}(x)}{\partial x_{i}}+\frac{\partial T^{\left(1\right)}(x,y)}{\partial y_{i}}\right)$$
(43)$$q_{k}^{\left(1\right)}(x,y)=-\kappa_{ik}(y)\left(\frac{\partial T^{\left(1\right)}(x,y)}{\partial x_{i}}+\frac{\partial T^{\left(2\right)}(x,y)}{\partial y_{i}}\right)$$

Substituting Equation (42) into (27), and then,
(44)$$\frac{\partial }{\partial y_{k}}\left\{\kappa_{ik}(y)\frac{\partial T^{\left(1\right)}(x,y)}{\partial y_{i}}\right\}=-\frac{\partial \kappa_{ik}(y)}{\partial y_{k}}\frac{\partial T^{\left(0\right)}(x)}{\partial x_{i}}$$

The separation of variables on the right–hand side of the above equation prompts us to write down the solution of *T*^(1)^ as,
(45)$$T^{\left(1\right)}(x,y)=L^{\prime }(x)+M^{k}(y)\frac{\partial T^{\left(0\right)}(x)}{\partial x_{k}}$$
where, *M*(**y**) is the local function of equilibrium problem to be solved likes *N^kl^*(**y**) and *L*(**y**), *L′*(**x**) is the homogeneous part of the solution.

Substituting Equation (45) into (44), the functions satisfy,
(46)$$\frac{\partial }{\partial y_{k}}\left\{\kappa_{lk}(y)\frac{\partial L^{\prime }(x)}{\partial y_{l}}\right\}=0$$
(47)$$\frac{\partial }{\partial y_{k}}\left\{\kappa_{lk}(y)\frac{\partial M^{i}(y)}{\partial y_{l}}\right\}=-\frac{\partial \kappa_{ik}(y)}{\partial y_{k}}$$

Substituting Equation (45) into (42), then,
(48)$$q_{k}^{\left(0\right)}(x,y)=-\left(\kappa_{ik}(y)+\kappa_{lk}(y)\frac{\partial M^{i}(y)}{\partial y_{l}}\right)\frac{\partial T^{\left(0\right)}(x)}{\partial x_{i}}$$

Substitute Equation (48) into (41), and subsequently average over the volume of the unit–cell. Thus,
(49)$$\frac{1}{\left|Y\right|}\int_{Y}\frac{\partial q_{k}^{\left(1\right)}(x,y)}{\partial y_{k}}dv-\frac{1}{\left|Y\right|}\int_{Y}\left(\kappa_{ik}(y)+\kappa_{lk}(y)\frac{\partial M^{i}(y)}{\partial y_{l}}\right)dv\frac{\partial T^{\left(0\right)}(x)}{\partial x_{i}\partial x_{k}}=0$$

Keeping the periodicity of the associated functions in mind and remembering to treat *x_i_* as a parameter as far as integration with respect to **y** is concerned results in an expression of the form,
(50)$$-{\bar{\kappa }}_{ik}\frac{\partial T^{\left(0\right)}(x)}{\partial x_{i}\partial x_{k}}=0$$With
(51)$${\bar{\kappa }}_{ik}=\frac{1}{\left|Y\right|}\int_{Y}\left(\kappa_{ik}(y)+\kappa_{lk}(y)\frac{\partial M^{i}(y)}{\partial y_{l}}\right)dv$$

The first integral vanishes as a consequence of the periodicity of $q_{k}^{\left(1\right)}$ and following the application of the divergence theorem. 

Prior to calculating the effective coefficients by Equations (39), (40) and (51), one must first determine the local functions $N_{m}^{kl}$; $L_{m}$; $M^{i}$ by Equations (34), (35) and (47) from the appropriate unit–cell problems.

### 3.3. Determination of Effective Coefficients

In this section, the equivalent performance of a graded composite considering the thermo–mechanical coupling effect is studied. Assuming that the graded composite, where the gradient layers are orthogonal to the axis *Ox*_3_ and the constituents of the structure have an arbitrary isotropic. The graded composite material satisfies the generalized periodicity. The local function of the composite material with generalized periodicity satisfies a set of partial differential equations, which is called the local problem on the periodic element. 

The coefficients of the equilibrium problem on a thermal mechanical coupling structure are rapidly oscillating functions. Then, Equations (34), (35) and (47) become,
(52)$$\frac{\partial }{\partial y_{3}}\left\{C_{i3k3}(y)\frac{\partial N_{n}^{kl}(y)}{\partial y_{3}}\right\}=-\frac{\partial C_{i3kl}(y)}{\partial y_{3}}$$
(53)$$\frac{\partial }{\partial y_{3}}\left\{C_{i3k3}(y)\frac{\partial L_{n}(y)}{\partial y_{3}}\right\}=\frac{\partial \beta_{i3}(y)}{\partial y_{3}}$$
(54)$$\frac{\partial }{\partial y_{3}}\left\{\kappa_{l3}(y)\frac{\partial M^{i}(y)}{\partial y_{3}}\right\}=-\frac{\partial \kappa_{i3}(y)}{\partial y_{3}}$$

Integrating once with respect to y_3_ yields,
(55)$$\frac{\partial N_{m}^{kl}}{\partial y_{3}}=-C_{m3i3}^{-1}C_{i3kl}+C_{m3i3}^{-1}A_{ikl}$$
(56)$$\frac{\partial L_{k}}{\partial y_{3}}=C_{k3i3}^{-1}\beta_{i3}+C_{k3i3}^{-1}B_{i}$$
(57)$$\frac{\partial M^{i}}{\partial y_{3}}=-\kappa_{33}^{-1}\kappa_{3i}+\kappa_{33}^{-1}C_{i}$$
where *A_ikl_*, *B_i_*, *C_i_* are constants of integration.

Based on the above relations, and integrating one more time with respect to *y*_3_, remembering at the same time that the functions $N_{m}^{kl}{(y}_{3})$, $L_{k}{(y}_{3})$ and $M^{i}(y_{3})$ are 1–periodic in *y*_3_, results in the following expression for the constants of integration,
(58)$$A_{ikl}={\langle C_{m3i3}^{-1}\rangle }^{-1}\langle C_{m3i3}^{-1}C_{i3kl}\rangle$$
(59)$$B_{i}=-{\langle C_{k3i3}^{-1}\rangle }^{-1}\langle C_{k3i3}^{-1}\beta_{i3}\rangle$$
(60)$$C_{i}={\langle \kappa_{33}^{-1}\rangle }^{-1}\langle \kappa_{33}^{-1}\kappa_{3i}\rangle$$

Substituting Equations (58)–(60) into Equations (55)–(57), then,
(61)$$\frac{\partial N_{m}^{kl}}{\partial y_{3}}=-C_{m3i3}^{-1}C_{i3kl}+C_{m3i3}^{-1}{\langle C_{m3i3}^{-1}\rangle }^{-1}\langle C_{m3i3}^{-1}C_{i3kl}\rangle$$
(62)$$\frac{\partial L_{k}}{\partial y_{3}}=C_{k3i3}^{-1}\beta_{i3}-C_{k3i3}^{-1}{\langle C_{k3i3}^{-1}\rangle }^{-1}\langle C_{k3i3}^{-1}\beta_{i3}\rangle$$
(63)$$\frac{\partial M^{i}}{\partial y_{3}}=-\kappa_{33}^{-1}\kappa_{3i}+\kappa_{33}^{-1}{\langle \kappa_{33}^{-1}\rangle }^{-1}\langle \kappa_{33}^{-1}\kappa_{3i}\rangle$$

Substituting Equations (61)–(63) into Equations (34), (35) and (47), we can get the effective elasticity/stiffness coefficients,
(64)$$C_{ijkl}^{H}=\langle C_{ijkl}\rangle -\langle C_{ijm3}C_{m3p3}^{-1}C_{p3kl}\rangle +\langle C_{ijm3}C_{m3p3}^{-1}\rangle {\langle C_{p3s3}^{-1}\rangle }^{-1}\langle C_{s3q3}^{-1}C_{q3kl}\rangle$$
(65)$$\beta_{ij}^{H}=\langle \beta_{ij}\rangle -\langle C_{ijm3}C_{m3p3}^{-1}\beta_{p3}\rangle +\langle C_{ijm3}C_{m3p3}^{-1}\rangle {\langle C_{p3s3}^{-1}\rangle }^{-1}\langle C_{s3q3}^{-1}\beta_{q3}\rangle$$
(66)$$\kappa_{ik}^{H}=\langle \kappa_{ik}\rangle -\langle \kappa_{3k}\kappa_{33}^{-1}\kappa_{3i}\rangle +\langle \kappa_{3k}\kappa_{33}^{-1}\rangle {\langle \kappa_{33}^{-1}\rangle }^{-1}\langle \kappa_{33}^{-1}\kappa_{3i}\rangle$$
where, *C*^H^, *β*^H^, *κ*^H^ are the effective elasticity tensor, the tensor related to effective thermal expansion tensor **α**^H^ and the effective thermal conductivity tensor, respectively. It is noted that **α**^H^ can be calculated by *β_ij_* = *C_ijkl_*·*α_kl_*.

For the isotropic materials, its elastic constant matrix **C**_6×6_, thermal expansion coefficient matrix **α**_3×3_, thermal conductivity coefficient matrix **κ**_3×3_ are given by,
(67)$$C_{6\times 6}=\left[\begin{array}{cccccc} C_{11} & C_{12} & C_{12} & 0 & 0 & 0\\ C_{12} & C_{11} & C_{12} & 0 & 0 & 0\\ C_{12} & C_{12} & C_{11} & 0 & 0 & 0\\ 0 & 0 & 0 & C_{66} & 0 & 0\\ 0 & 0 & 0 & 0 & C_{66} & 0\\ 0 & 0 & 0 & 0 & 0 & C_{66} \end{array}\right],\;\alpha_{{}_{3\times 3}}=\left(\begin{array}{ccc} \alpha_{11} & 0 & 0\\ 0 & \alpha_{11} & 0\\ 0 & 0 & \alpha_{11} \end{array}\right),\;\kappa_{{}_{3\times 3}}=\left(\begin{array}{ccc} \kappa_{11} & 0 & 0\\ 0 & \kappa_{11} & 0\\ 0 & 0 & \kappa_{11} \end{array}\right)$$
where, *C*_11_ = E(1 − *v*)/[(1 + *v*)(1 − 2*v*)], *C*_12_ = E*v*/[(1 + *v*)(1 − 2*v*)], *C*_66_ = (*C*_11_ − *C*_12_)/2.

Thus, the analytical formulas of the effective properties for FGMs are assumed to be made by a mixture of two isotropic elastic constituents that are derived and we can obtain the following closed–form formulas,

(1) For the effective elastic tensor’s components,
(68)$$\begin{array}{l} C_{11}^{H}=\langle C_{11}\rangle -\langle C_{12}^{2}C_{11}^{-1}\rangle +{\langle C_{12}C_{11}^{-1}\rangle }^{2}{\langle C_{11}^{-1}\rangle }^{-1},\\ C_{12}^{H}=\langle C_{12}\rangle -\langle C_{12}^{2}C_{11}^{-1}\rangle +{\langle C_{12}C_{11}^{-1}\rangle }^{2}{\langle C_{11}^{-1}\rangle }^{-1},\\ C_{13}^{H}=C_{23}^{H}=\langle C_{12}C_{11}^{-1}\rangle {\langle C_{11}^{-1}\rangle }^{-1},\\ C_{33}^{H}={\langle C_{11}^{-1}\rangle }^{-1},\\ C_{44}^{H}=C_{55}^{H}=\frac{1}{2}{\langle {\left(C_{11}-C_{12}\right)}^{-1}\rangle }^{-1},\\ C_{66}^{H}=\frac{1}{2}\langle C_{11}^{H}-C_{12}^{H}\rangle \end{array}$$
where, <·> is the average. It can be discovered that the FGMs are transversely isotropic and have 5 independent constants ($C_{66}^{H}$ is not an independent constant), the matrix is symmetric.

(2) For the effective thermal expansion tensor’s components, due to
(69)$$\begin{array}{l} \beta_{11}^{H}=\beta_{22}^{H}=\langle \beta_{11}\rangle -\langle C_{12}C_{11}^{-1}\beta_{11}\rangle +\langle C_{12}C_{11}^{-1}\rangle {\langle C_{11}^{-1}\rangle }^{-1}\langle C_{11}^{-1}\beta_{11}\rangle ,\\ \beta_{33}^{H}={\langle C_{11}^{-1}\rangle }^{-1}\langle C_{11}^{-1}\beta_{11}\rangle \end{array}$$
thus, we can calculate the effective thermal expansion tensor’s components *α_kl_* in terms of the *β_ij_* = *C_ijkl_*·*α_kl_*.

(3) For the effective thermal conductivity tensor’s components,
(70)$$\kappa_{11}^{H}=\kappa_{22}^{H}=\kappa_{33}^{H}=\langle \kappa_{11}\rangle$$

## 4. Numerical Results and Discussion

### 4.1. Validation of the Present Method

This section mainly verifies the validity of the analytical method based on the asymptotic homogenization method proposed in this paper. We compare the calculation results of this method with the theoretical and numerical results respectively, and the correctness is verified.

Firstly, the composites we studied are FGMs composed of Mo and ZrC. Due to the lack of relevant experimental data, the results of the present method are compared with the classical method of Mori–Tanaka for predicting the effective coefficients of composites. The relevant material parameters at room temperature are given in Table 1. The material properties calculated by the methods of present and Mori–Tanaka are plotted as a function of the volume fraction of ZrC in Figure 4. It is observed that for this special material combination, the homogenized material properties are almost the same as those obtained by Mori–Tanaka method. Compared with Mori–Tanaka method, $C_{11}^{H}$ and $C_{66}^{H}$ calculated by the analytical method are larger, while $C_{12}^{H}$ is smaller. On the whole, however, the analytical results of the AHM are in good agreement with those of the Mori–Tanaka method.

On the other hand, the equivalent coefficient of Al/SiC fiber reinforced metal matrix composites was calculated. The elastic modulus of Al and SiC are 70.0 and 450.0 Gpa, and Poisson’s ratio are 0.30, 0.17, respectively. The homogenized results obtained using the present method are compared to the results obtained using the AHM on Abaqus [15], as shown in Figure 5. Three different fiber volume fractions were investigated: 20%, 35% and 55%. The results obtained by [10] are also shown in Figure 5. It is possible to observe that the homogenized coefficients obtained using the present method are practically the same as those obtained in [15]. For all the results the differences are less than 1.5%.

In summary, the effectiveness of our present method has been proved. Therefore, the thermo–mechanical properties of FGMs with different gradient changes will be discussed according to this method.

### 4.2. Effective Elastic Tensor’s Components

In Figure 6, the effective elastic tensor’s components perturbation along the gradient direction are plotted by selecting distinct component ratio coefficient *c*_0_ while fixing the gradient index as *m* = 1.0. Assume that the component ratio coefficient varies from *c*_0_ = 0 to *c*_0_ = 1.0. For FGMs (*c*_0_ ≠ 0), the variation of effective elastic tensor’s components depends on the gradient distribution of the component volume fraction, and presents a nonlinear change trend. With the increase of component ratio coefficient *c*_0_, the performance of the bottom material remains unchanged because there is only the material Mo when *z* = 0; However, the material properties in other places except the bottom are changing more and more sharply, and the difference of material properties between the top and bottom is also increasing. This indicates that the larger the component ratio coefficient *c*_0_, the more obvious the difference of local material properties of FGMs; the volume fraction of the material phase ZrC increases with the increase of *c*_0_, which affects the properties of the material to a great extent.

In Figure 7, when the component ratio coefficient *c*_0_ = 1.0 [*V*_ZrC_:*V*_Mo_ = 0.5:0.5], the influences of gradient index *m* on the perturbation of the effective elastic tensor’s component along the gradient direction is shown. It is assumed that the gradient index of volume fraction varies from *m* = 0 to *m* = 10.0. For FGMs (*m* ≠ 0), the variation of the effective elastic tensor’s components depends on the gradient distribution of volume fraction while exhibiting similar or anti–similar trends with that shown in Figure 1b. With the increase of gradient index *m*, the difference of material properties on the top and bottom surfaces remains constant, because the materials of these location do not vary with the change of gradient index *m*, and they are only related to the volume ratio *c*_0_; However, with the increase of gradient index *m*, the variation of material properties near the top and bottom surfaces is slower and slower, but the variation of material properties is more severe when approaching the middle surface, which indicates that the inhomogeneity of material phase gradient changes leads to the inhomogeneity of material property changes, mainly determined by the different volume fraction at the current position *z*/*h* caused by gradient index *m*. The distribution of the material properties of FGMs along the gradient direction is asymmetric and with a turning point when *z*/*h* = 0 which means *V*_ZrC_:*V*_Mo_ = 0.5:0.5 in this section.

### 4.3. Effective Thermal Expansion Tensor’s Components

The graphs present changes in the effective thermal expansion coefficient tensor’s components depending on the component ratio coefficient *c*_0_ when fixing the gradient index as *m* = 1.0 (Figure 8) and the gradient index *m* when fixing the component ratio coefficient as *c*_0_ = 1.0 (Figure 9).

As can be seen from Figure 8, for FGMs (*c*_0_ ≠ 0), the change of the effective thermal expansion tensor’s components depends on the gradient distribution of the volume fraction of the component, and presents a nonlinear change trend. The effective thermal expansion tensor’s components increase with the increase of ZrC volume fraction, which seems to indicate that FGMs are easier to deform after thermal shock than Mo, and the thermal shock resistance of FGMs is not optimized compared with Mo. At the same time, with the increase of the component ratio coefficient *c*_0_, the effective performance of the bottom surface remains unchanged, but the top surface is larger and larger, i.e., the difference between the effective thermal expansion tensor’s components of the top surface and the bottom surface is larger and larger, and the overall difference of FGMs is obvious. When the structural FGMs change linearly, the nonlinear changes of the effective tensor’s components of elasticity and thermal expansion are observed, which indicates that the gradient change not only affects the volume ratio of the component phases, but also leads to the influence of other aspects on the overall material performance after the two materials are compounded. Thus, considering the microstructures of FGMs is vital when studying their effective properties.

In Figure 9, the change of the effective thermal expansion tensor’s components with the gradient position exhibits a trend similar to FGMs (*m* ≠ 0) shown in Figure 1b. With the increase of the gradient index *m*, the effective thermal expansion tensor’s components are the same as the effective elastic tensor’s components, and the performance difference between the top surface and the bottom surface remains unchanged, because the materials at these positions do not change with the change of the gradient index *m*, but only relate to the volume ratio *c*_0_; However, with the increase of gradient index *m*, the material properties near the top and bottom surface change more slowly, but the material properties change more and more sharply near the middle surface. The effective thermal expansion tensor’s components of FGMs is asymmetrically distributed along the gradient direction, and there is a turning point when *z*/*h* = 0, i.e., *V*_ZrC_:*V*_Mo_ = 0.5:0.5.

### 4.4. Effective Thermal Conductivity Tensor’s Components

The graphs present the changes in the thermal conductivity tensor’s components depending on the component ratio coefficient *c*_0_ when the gradient index *m* = 1.0 (Figure 10a) and the gradient index *c*_0_ when the component ratio coefficient *c*_0_ = 1.0 (Figure 10b).

From Figure 10, the variation of thermal conductivity tensor’s components depends on the gradient distribution of the component volume fraction, and presents a linear change trend. The thermal conductivity tensor’s components of the gradient material decrease with the increase of ZrC volume fraction, which indicates that the internal heat transfer efficiency of the material is decreasing. When the component ratio coefficient *c*_0_ increases, this decrease is more obvious in Figure 10a; with the increase of the gradient index *m*, the range of the overall thermal conductivity does not change, but there are local differences, and the smaller *m*, the smaller the local differences in Figure 10b.

Considering the effective tensor’s components of thermal expansion and thermal conductivity, the change of the effective performance tensor’s components is closely related to the volume fraction of the component phase. With the increase of ZrC volume fraction, the thermal expansion tensor’s components of FGMs increases, i.e., FGMs are more likely to deform when subjected to thermal shock. However, the thermal conductivity tensor’s components are decreasing, that is to say, it is more difficult to conduct heat in FGMs. However, in general, the deformation effect of FGMs caused by the increase of thermal expansion tensor’s components is less than that caused by the decrease of thermal conductivity tensor’s components, which means that under the same heating conditions, FGMs have smaller thermal expansion degree and better thermal shock resistance than single metal material Mo.

## 5. Conclusions

In summary, a methodology to find the local problems and the effective thermo–mechanical coefficients equations for structures with generalized periodicity is presented based on the method of AHM. Following the proposal, the homogenized problem for FGMs is derived, the relationship between the structure and properties of Mo/ZrC FGMs is studied. The constituent ratio and the property gradation profiles of Mo/ZrC FGMs are described by design volume fraction. From the calculation results, it can be observed that the component ratio coefficient and gradient index have important influences on the thermo–mechanical properties of FGMs. The component ratio coefficient affects the overall performance difference of FGMs, while the gradient index affects the local performance difference. This work opens up the possibility to predict the changing trend of FGMs performance with different gradient structures, and provides guidance for further innovation and unique design of new FGMs structures.

## Figures and Tables

**Figure 1 materials-15-03073-f001:**
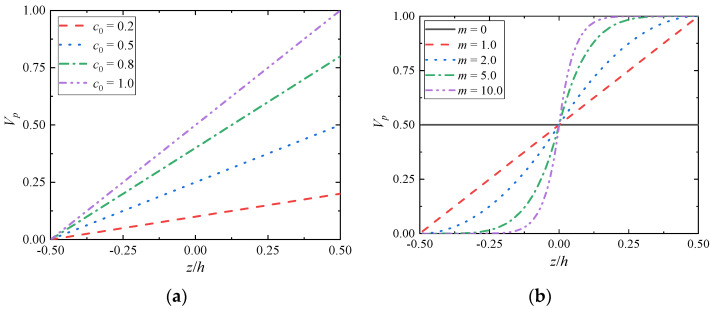
Variation of volume fraction *V_p_* vs. FGMs structure: (**a**) the variational material component ratio coefficients *c*_0_ (*m* = 1.0); (**b**) the variational gradient indexes *m* (*c*_0_ = 1.0).

**Figure 2 materials-15-03073-f002:**
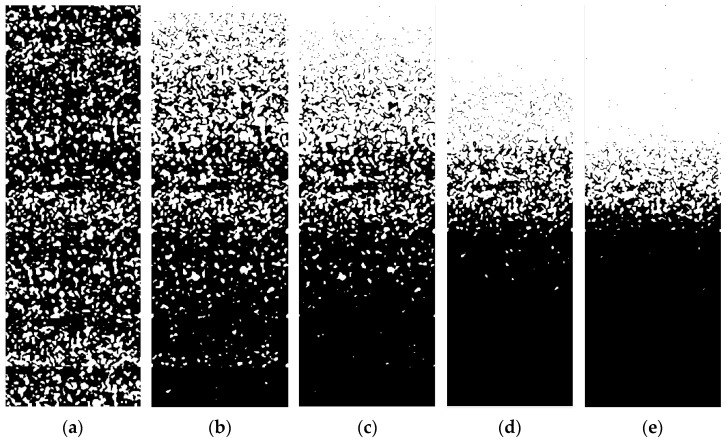
Establishment of the gradient model with (**a**) *m* = 0; (**b**) *m* = 1.0; (**c**) *m* = 2.0; (**d**) *m* = 5.0; (**e**) *m* = 10.0 (*c*_0_ = 1.0).

**Figure 3 materials-15-03073-f003:**
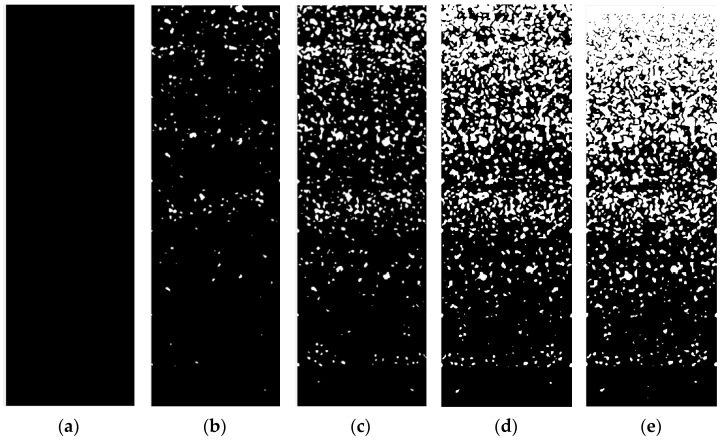
Establishment of the gradient model with (**a**) *c*_0_ = 0; (**b**) *c*_0_ = 0.2; (**c**) *c*_0_ = 0.5; (**d**) *c*_0_ = 0.8; (**e**) *c*_0_ = 1.0 (*m* = 1.0).

**Figure 4 materials-15-03073-f004:**
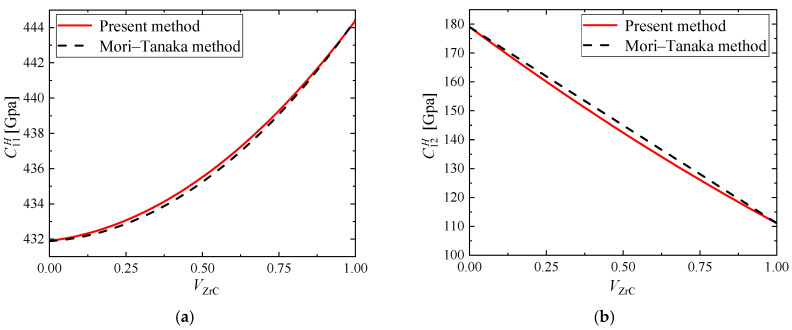
Comparison of two methods for effective elastic tensor’s components (**a**) $C_{11}^{H}$; (**b**) $C_{12}^{H}$; (**c**) $C_{66}^{H}$.

**Figure 5 materials-15-03073-f005:**
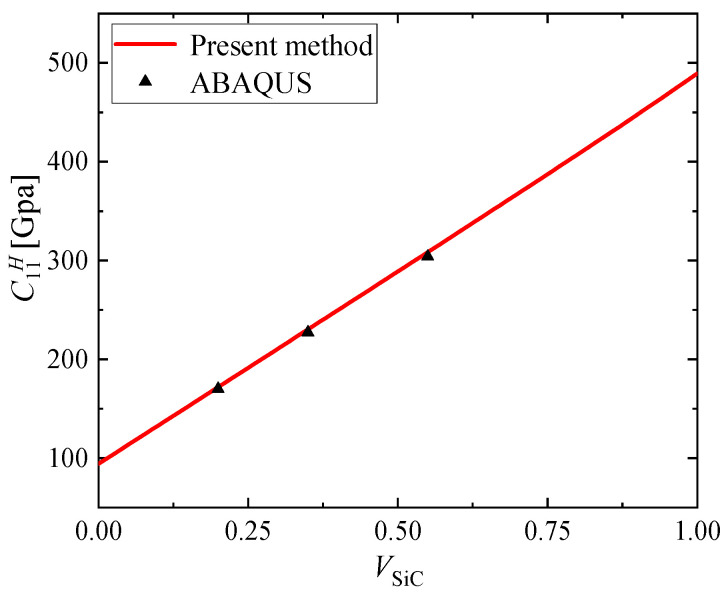
$C_{11}^{H}$ obtained using the present method and the AHM on Abaqus [15].

**Figure 6 materials-15-03073-f006:**
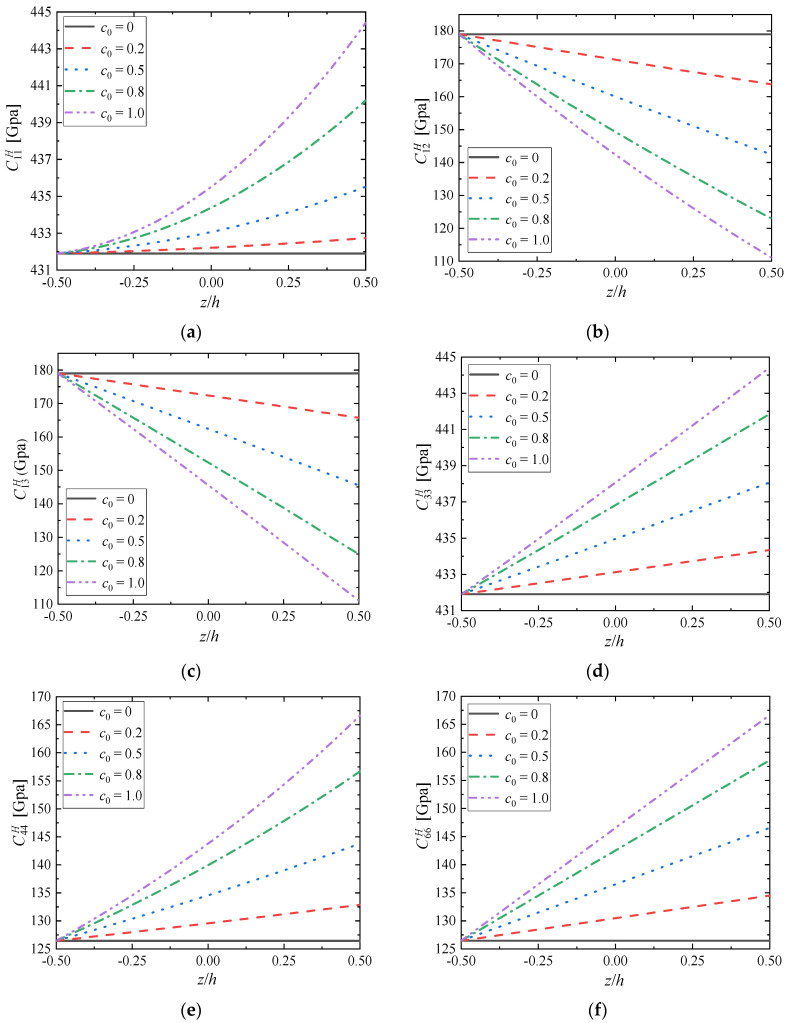
Distributions of (**a**) $C_{11}^{H}$; (**b**) $C_{12}^{H}$; (**c**) $C_{13}^{H}$; (**d**) $C_{33}^{H}$; (**e**) $C_{44}^{H}$; (**f**) $C_{66}^{H}$ along the gradient direction when *m* = 1.0.

**Figure 7 materials-15-03073-f007:**
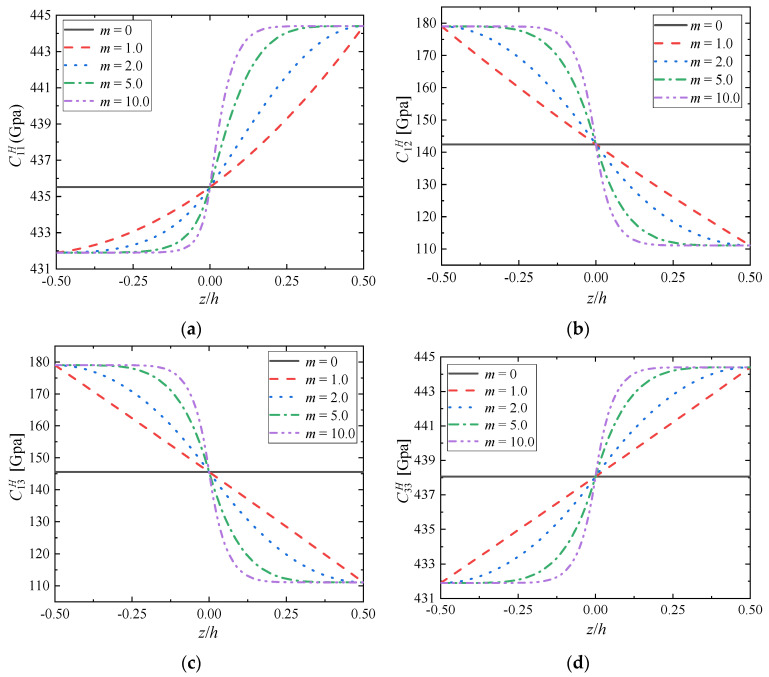
Distributions of (**a**) $C_{11}^{H}$; (**b**) $C_{12}^{H}$; (**c**) $C_{13}^{H}$; (**d**) $C_{33}^{H}$; (**e**) $C_{44}^{H}$; (**f**) $C_{66}^{H}$ along the gradient direction when *c*_0_ = 1.0.

**Figure 8 materials-15-03073-f008:**
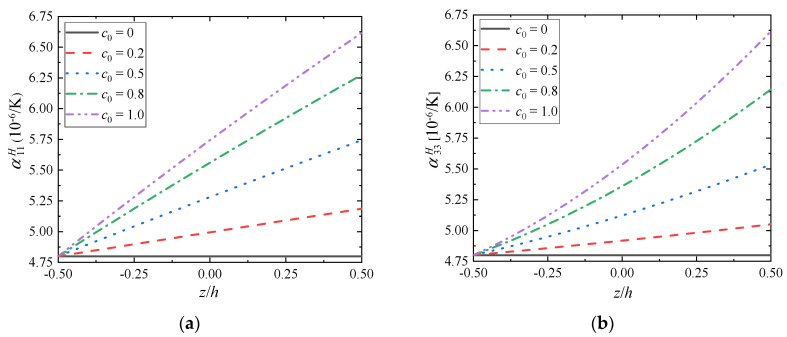
Distributions of (**a**) $\alpha_{11}^{H}$ and (**b**) $\alpha_{33}^{H}$ along the gradient direction (*m* = 1.0).

**Figure 9 materials-15-03073-f009:**
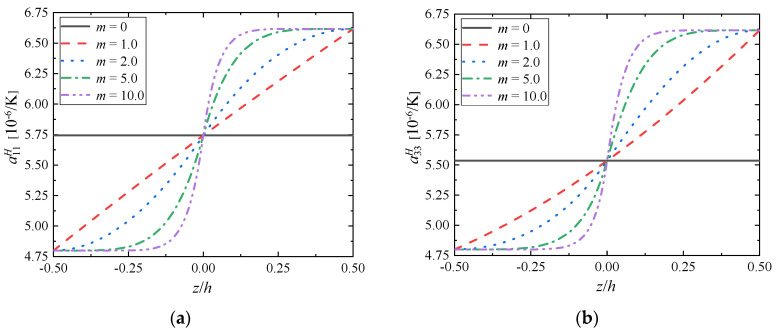
Distributions of (**a**) $\alpha_{11}^{H}$ and (**b**)$\;\alpha_{33}^{H}$ along the gradient direction (*c*_0_ = 1.0).

**Figure 10 materials-15-03073-f010:**
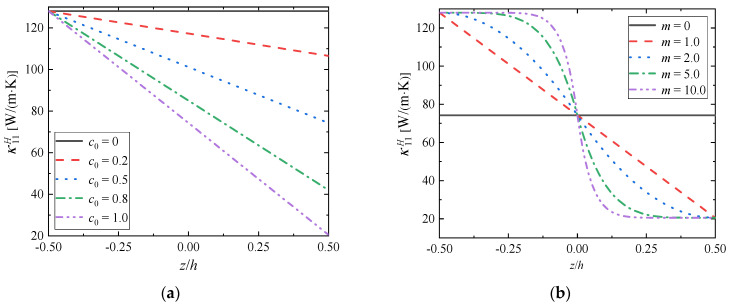
Distributions of $\;\kappa_{11}^{H}$ along the gradient direction when (**a**) *m* = 1.0; (**b**) *c*_0_ = 1.0.

**Table 1 materials-15-03073-t001:** The elastic modulus *E*, Poisson’s ratio *v*, thermal expansion coefficient *α* and thermal conductivity coefficient *κ* of materials at room temperature (25 °C).

Materials	*E* [Gpa]	*v*	*α* [10^−6^/K]	*κ* [W/(m·K)]
ZrC	400 [10]	0.20 [62]	6.7 [62]	20.52 [63]
Mo	327 [64]	0.293 [65]	4.8 [64]	128.0 [64]

## Data Availability

All data generated or analysed during this study are included in this published article.

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
