# Peer review of "Analytical Solution of Thermo–Mechanical Properties of Functionally Graded Materials by Asymptotic Homogenization Method"

_materials, 2022, doi:10.3390/ma15093073_

Round 1

Reviewer 1 Report

The paper deals with the asymptotic and numerical analysis of a boundary value problem in a bounded domain with inhomogeneous microstructure. The authors used formal asymptotic analysis to simplify numerical calculations, but one can find no verifications of the applied approach. The authors choose an asymptotic expansion in integer powers of epsilon without any comments. Also it should be noted that the authors leave the asymptotics (30) without any arguments.

Reviewer 2 Report

The manuscript by the authors presents an analytical solution for the effective properties of functionally graded materials (FGMs) with generalized periodicity using the asymptotic homogenization method (AHM). The paper is a good and novel study on this subject. However, it gives a sloppy impression, although the results could be interesting. Perhaps due to the presentation the reader cannot fully appreciate their work. At various points the paper needs to be revised:

The in-text references are not correctly shown in a lot of places, which are ‘Error! Reference source not found’.

There are a lot of repeated figures, Figs 1-3 repeated 3 times in pages 5-9.

The figure numbers are not correct/consistent, starting from Fig. 1 in page 16.

Some of the figures have a quite poor quality, e.g. Fig. 2a in page 17.

After fixing the above issues, the in-text Figure numbers should be corrected as well.

Equation numbers are not correct in a lot of places, e.g. repeated 20-21 in page 11, the numbers are total chaos in page 12, page 15.

It would be very nice if a Nomenclature illustrating all the parameters appeared in the manuscript is added, for papers focusing on theory.

The current introduction could briefly include the applications of FGMs in different areas, such as in electronic packaging, automobile and aerospace industries (10.1016/j.compscitech.2005.09.003; 10.1016/j.compstruct.2015.08.113; 10.1016/j.compositesb.2015.07.018).

The model can be better presented if a schematic illustration of the parameters (e.g. Eqs. 1-2) can be shown in the format of ‘Figure’.

Reviewer 3 Report

In this article, the authors studied general mathematical model for functionally graded heterogeneous equilibrium boundary value problems. A methodology to find the local problems and the effective properties of functionally graded materials with generalized periodicity is presented, using the asymptotic homogenization method. The results show that the designed structure profiles have great influence on the effective properties of the present inhomogeneous heterogeneous models.

The manuscript appears to be mathematically correct and the utilized methods and the proposed models are interesting. I recommend the paper for publication, however, there are some concerns, comments and suggestion should be addressed before publication:

  1. There are grammar and typographic errors. Please correct these errors and further improve the language.
  2. Last paragraph of introduction should be removed.
  3. In this paper, how do the authors use the particle size?
  4. The novelty of this work must be presented in Conclusion section very clear.
  5. In the Introduction section, it is suggested that some literature published in recent two years (2020-2021) should be included and also, some of the following literature concerned missing and these should be appropriately cited.

State-of-the-art review of fabrication, application, and mechanical properties of functionally graded porous nanocomposite materials

https://doi.org/10.1016/j.tws.2020.106841;

  1. The results and figures are appropriate however; author should add more physical explanation for the observed results.
  2. Need references. And author should add references if they did not drive and investigated those Eqs. by them self.
  3. Numerical simulations sound very comprehensive. Are there any assumptions behind this study that could restrict its practical application? Assumptions can be highlighted.
  4. Discuss the imposition of boundary conditions.

Round 2

Reviewer 1 Report

The paper can be published in the present form.